neuroscience/molecular biology/cellular biology

neurovascular, translational control, brain development, neurons, endothelial cells

**Author for correspondence:**
Christos G. Gkogkas
e-mail: cgkogkas@imbb.forth.gr

# Translational control in neurovascular brain development

Kleanthi Chalkiadaki, Elpida Statoulla, Maria Markou, Sofia Bellou, Eleni Bagli, Theodore Fotsis, Carol Murphy and Christos G. Gkogkas

Division of Biomedical Research, Institute of Molecular Biology and Biotechnology, Foundation for Research and Technology-Hellas, University Campus, 45110 Ioannina, Greece

KC, 0000-0001-7951-8942; ES, 0000-0003-3419-3253; CGG, 0000-0001-6281-3419

The human brain carries out complex tasks and higher functions and is crucial for organismal survival, as it senses both intrinsic and extrinsic environments. Proper brain development relies on the orchestrated development of different precursor cells, which will give rise to the plethora of mature brain cell-types. Within this process, neuronal cells develop closely to and in coordination with vascular cells (endothelial cells (ECs), pericytes) in a bilateral communication process that relies on neuronal activity, attractive or repulsive guidance cues for both cell types and on tight-regulation of gene expression. Translational control is a master regulator of the gene-expression pathway and in particular for neuronal and ECs, it can be localized in developmentally relevant (axon growth cone, endothelial tip cell) and mature compartments (synapses, axons). Herein, we will review mechanisms of translational control relevant to brain development in neurons and ECs in health and disease.

## 1. Coordinated neurovascular development in the central nervous system

The central nervous system (CNS) comprises the brain and spinal cord and contains various centres that integrate information from the entire body, coordinating a range of higher functions such as movement, speech and cognition. Herein, we will focus on neurovascular development in the brain. The formation of the CNS begins during early embryogenesis. In mice, this process initiates with neurulation at approximately embryonic day 7.5 (E7.5), whereby the neural plate establishes in the dorsal ectoderm and subsequently folds into the neural tube. Subsequently, the neural tube closes at approximately E9.5 and

**Figure 1.** Coordinated neurovascular development in the CNS adapted from [1–3]. (*a*) The first step in CNS vascularization is the invasion of the angioblasts in the neural tube, followed by PNVP and INVP formation, (*b*) depiction of E9.5 until E18.5 of embryogenesis in the CNS, whereby key brain cell-types emerge. OLP, oligodendrocyte precursors; NSC/NPC, neural stem cells/ neural progenitor cells, (*c*) common cellular structure shared by axonal growth and endothelial tip cells; red–blue: guidance cues and (*d*) the NVU: neurons communicate with the vascular cells via the astrocytes, while pericytes support the endothelial cells.

then becomes regionalized, differentiating into the primary brain vesicles: forebrain, midbrain and hindbrain and the caudal spinal cord. At this stage, the neural tube is not vascularized. Upon closure of the neural tube, endothelial progenitor cells (angioblasts), emanating from the presomitic mesoderm will instigate the first step in CNS vascularization, which is the formation of the perineural vascular plexus (PNVP), a primitive vascular network [1–4] (figure 1*a*). A cardinal pro-angiogenic signal originating from the neural tube that initiates CNS vasculogenesis is vascular endothelial growth factor A (VEGF-A) [5,6].

At E10.5, a secondary wave of sprouting angiogenesis from PNVP, invading radially from basal to the apical neural tube, leads to the formation of the intraneural vascular plexus (INVP) [7]. Concomitant with PNVP–INVP formation, dorsoventral patterning of neural tube progenitors is established [8]. Neuroepithelial cells transform into radial glia cells (RGCs), which are neural progenitors. Within the INVP, vascular patterning takes place via angiogenic sprouts, which co-develop with RGC fibres [3,9]. When vessels reach the ventricle, new branches emerge, they reverse their direction towards the pia and ultimately anastomose to form a rich capillary plexus called the periventricular vascular plexus (PVP) [9]. Progenitor proliferation, differentiation and ultimately differentiation and migration will give rise to the cell types of the brain: neurons and glia. Strikingly, inhibitory but not excitatory cortical neurons require vascular support for proper neurogenesis (figure 1*b*). Neural progenitor cells (NPCs) from the ventral telencephalon, which give rise to inhibitory neurons; (I) require the association with blood vessels, while NPCs from the dorsal region, which give rise to excitatory neurons; (E) do not [10]. Initially, both NPC populations (E/I) are associated with the pial basement membrane, but from E14.5, ventral telencephalic NPCs switch to periventricular blood vessels [10]. This event is required to promote NPC division and neocortical interneuron neurogenesis.

At the tip of the growing neuronal axon, AGC (axonal growth cones) extend fan-like lamellipodial and long, finger-like filopodial protrusions that sense the local microenvironment for guidance cues to steer the developing axon [4]. Sprouting blood vessels are led by ECs that resemble these axonal

growth cones in cellular appearance and function, exhibiting similar lamellipodia and filopodia structures [4]. These cells have been named 'endothelial tip cells' and are key structures in the pathfinding of developing, newly forming blood vessels [4]. AGC and EC tip cells sense attractive and repulsive guidance cues in the local tissue microenvironment, driving axon guidance to form a synapse and induce angiogenesis, respectively [11] (figure 1c). Common molecular cues termed angioneurins [12] are known to steer both for AGC and EC tip cells, such as axonal guidance molecule families, including Netrins, Semaphorins, Ephrins, Slits and their receptors, morphogens such as Wnts (wingless-type proteins), Shh (Sonic Hedgehog) and BMP (Bone Morphogenetic Protein) and classical angiogenic factors like VEGF-A, Fibroblast growth factor 2 (FGF-2) and vessel-derived factors like Endothelin-3 and Artemin and its receptor GFRα3 (GDNF—glial cell line-derived neurotrophic factor—family receptor α-3). Furthermore, AGC and EC tip cells rely on mRNA transport and localization [13] at distal sites to support AGC guidance and EC tip cell pathfinding by compartmentalizing gene expression at these sites. Undoubtedly, neuroscience and endothelial cell biology research have made great advances regarding our understanding of cue-mediated growth and mRNA localization in neuronal and endothelial cells, and these are extensively reviewed in [1–3].

The formation of the blood–brain barrier (BBB) takes place while blood vessels develop within the CNS and thus vessels can tightly regulate the movement of ions, molecules and cells between the blood and the brain [14]. BBB function relies on the unique properties of brain ECs compared with peripheral ECs in other tissues [14]. First, brain ECs form tight cell–cell junctions, a unique architecture that significantly restricts paracellular solute flux. Second, transport of solutes in vesicles is limited by the extremely low rates of transcytosis in brain ECs. Third, brain ECs lack fenestration and express specific molecular transporters. Thus, only regulated, selective, molecular transport is permitted via the BBB. ECs recruit pericytes to the developing vasculature as soon as vessels enter the neural tube, which enables the vasculature to acquire BBB properties. Pericytes are a subtype of mural cells, and they incompletely cover the endothelial walls of small vessels in the brain. Vascular smooth muscle cells (VSMCs) are the second type of mural cells and surround the large vessels [15]. With the advent of small conditional RNA sequence scRNAseq (scRNAseq), there is recent evidence that brain vasculature comprises heterogeneous cell-types, which are transcriptionally diversified and subspecialized compared with other vascularized organs [16,17]. A relatively recent concept in neuroscience concerns the cellular ensemble formed by ECs, mural cells, astrocytes and neurons, and is named the neurovascular unit (NVU) (figure 1d). The structure/function of the adult NVU is well described, in stark contrast to the poorly studied NVU in all three stages of the developing brain; embryonic, fetal and early postnatal [7].

The tight interplay between neurons and vascular cells within the neurovascular unit at the molecular, cellular and physiological level coordinates and dictates CNS development, homeostasis and function.

## 2. Translational control of gene expression

Cells respond to internal or external stimuli by changing their function or phenotype, in order to adapt to the new conditions/environmental challenges, and ultimately survive. This process depends on the regulation of gene expression at multiple levels. Transcription is the first step in the gene-expression pathway and while it can be rate limiting for the final protein product [18], it is also a time-consuming and energy-demanding process [19]. Regulation at the level of mRNA translation is in several instances a better predictor than transcription for protein abundance [20,21]. On the other hand, translational control provides an additional level of gene expression control, by shaping not only the abundance but also the spatio-temporal expression of proteins [22]. Translational control is defined as the sum of regulatory events dictating the amount of protein produced per mRNA, prior to post-translational regulation, such as modifications and degradation [23]. Translational control can be distinguished into global, affecting protein synthesis rate for most mRNAs and selective, whereby protein synthesis rate is preferentially regulated for subsets of mRNAs. Translational control provides the organism with a fast and precise mechanism of adaptation [19], by regulating small changes in the protein levels, using pre-existing mRNAs, which are immediately available in the cell [24].

Translation is a highly conserved mechanism among species and is divided into three stages: initiation, elongation and termination/recycling [25]. In eukaryotic cells, initiation is a complex process, which is considered the rate-limiting step and is, therefore, the main stage of translational control (figure 2a). Mature mRNAs possess a cap structure at their 5'-end, the m7GpppN, where m is a methyl group and N is any nucleotide. The addition of the cap structure is an indispensable post-

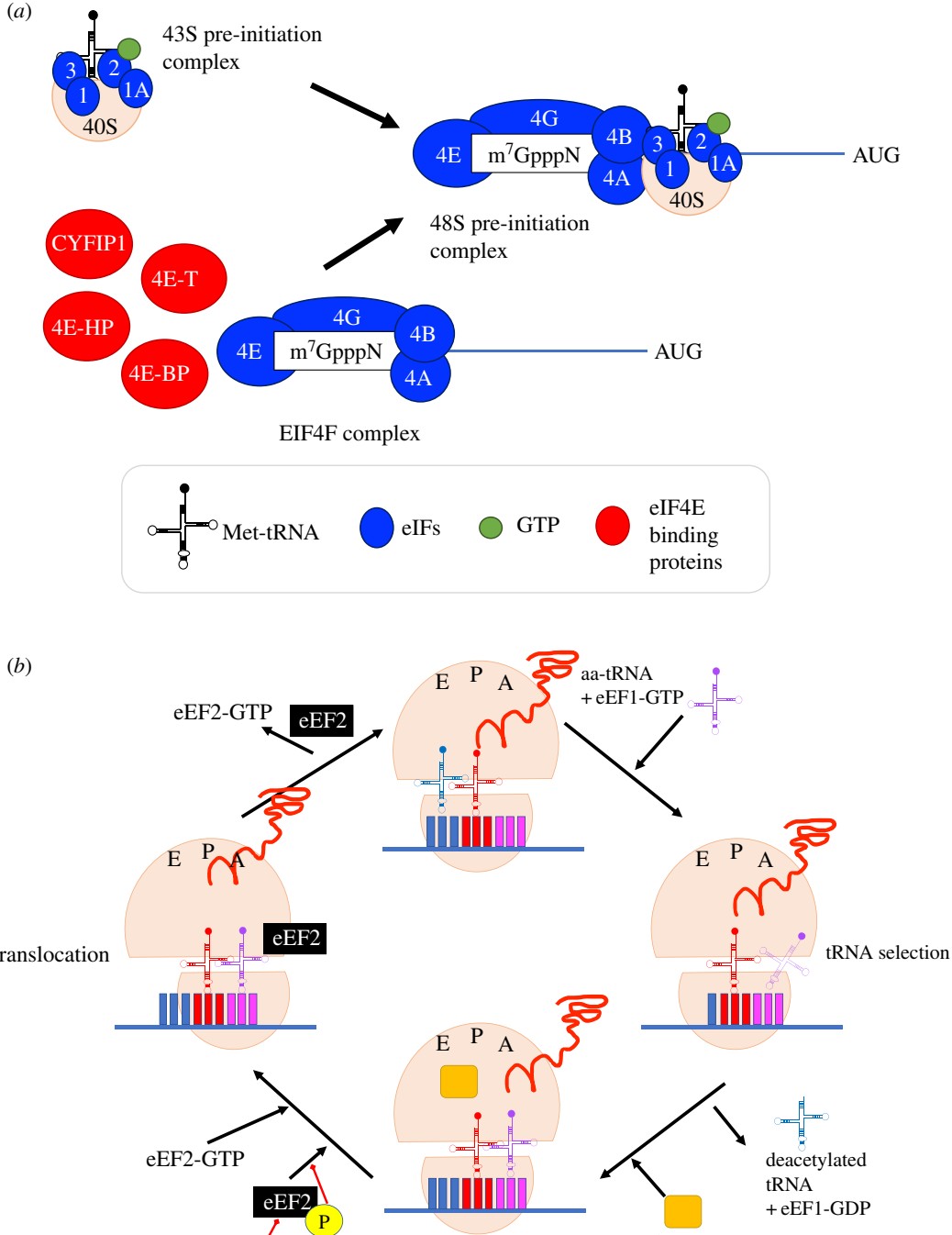

**Figure 2.** Translation initiation and elongation adapted from [26,27]. (*a*) Translation initiation, depicting the formation of the eIF4F and 43S preinitiation complexes. eIF4F complex comprises eIF4E, eIF4G, eIF4A and eIF4B. Interaction of eIF4F and 43S PIC gives the 48S PIC. Recognition of the initiation codon by the eIF4F complex leads to the 60S ribosomal subunit addition and formation of the elongating 80S ribosome, which signals the transition from initiation to elongation. 4E-BPs and CYFIP1 compete with each other for binding to eIF4E and act as cap-dependent translation regulators at the initiation step. 4E-T and 4E-HP bind to eIF4E and exert additional translational control in specific mRNAs. (*b*) Translation elongation comprises tRNA selection, peptidyl transfer and translocation. An aa-tRNA is initially recruited to the A site of the ribosome, followed by its pairing with the cognate codon and hydrolysis of the eEF1-bound GTP. Then the A and P site tRNAs interact, peptide bond formation occurs and finally, the polypeptide chain is transferred to the aa-tRNA. eIF5A1 supports the formation of certain peptide bonds [27]. At the final step of elongation, eEF2 enters the A site of the ribosome, induces a ribosomal conformation change and leads to translocation of the ribosome. eEF2 kinase act as a regulator of elongation, by inhibiting eEF2 through phosphorylation. Recognition of the stop codon signals the end of elongation and subsequent termination. Ribosomal conformation changes during elongation are shown by displacement of tRNA molecules.

transcriptional modification of newly synthesized mRNAs [28]. There are two types of initiation of translation: cap-dependent and cap-independent. Cap-dependent initiation requires the binding of eIF4E (eukaryotic initiation factor 4E) to the cap structure of the translated mRNA [29,30] (figure 2). Briefly, cap-dependent initiation begins with the formation of the 43S PIC (preinitiation complex) at the 5′-end of the mRNA. The Met-tRNA (initiator tRNA), together with eIF2 (eukaryotic initiation factor 2) and a GTP (Guanosine-5′-triphosphate) molecule bind to the 40S ribosomal subunit [31], and with the addition of the eIF4F complex, which consists of eIF4E, eIF4G, eIF4A and eIF4B [31], form the 48S PIC [32]. During initiation, the eIF4F complex scans the 5′ mRNA UTR (untranslated region) from 5′ to 3′ and, upon recognition of the AUG (initiation codon), the 60S ribosomal subunit is added in an eIF5-dependent step, followed by GTP hydrolysis (80S initiation complex formation) [31]. Once the 80S complex is assembled, translation enters into its elongation phase [32].

Global translation control can occur at the stage of the 43S PIC formation. Phosphorylation of eIF2α at Ser51 inhibits the exchange of Guanosine-5′-diphosphate (GDP) to GTP, and therefore, the formation of the 43S PIC formation complex, preventing the initiation of protein synthesis [33]. Remarkably, at the same time, eIF2α Ser51 phosphorylation stimulates Activating Transcription Factor 4 (ATF4) translation via upstream open reading frames (uORFs) [34]. Over 40% of eukaryotic mRNAs possess uORFs, thus phospho-eIF2α translational control may play a pervasive regulatory role [35]. In eukaryotic organisms, mature mRNAs possess an additional structure, a post-translational modification, the 3′ end poly-A tail, consisting of 50–300 adenylates, which interacts with Poly-A Binding Protein (PABP) [31]. PABP binds to eIF4G and forms a closed loop that enhances translation [36]. This interaction serves as a regulatory mechanism of protein synthesis at the initiation step. In parallel, the activity of eIF4E, and therefore, cap-dependent translation, is mainly regulated by the 4E-BPs (eIF4E-binding proteins) and the MAP -(mitogen-activated protein-kinase-interacting kinases (MNKs). During the initiation step, 4E-BPs can act as inhibitors of translation as they compete with eIF4G for binding to eIF4E [37,38]. 4E-BPs and eIF4G share the eIF4E-binding motif YXXXXLΦ, where X is any amino acid and Φ is a hydrophobic residue, which interacts with the convex dorsal surface of eIF4E [39]. Their activity depends on their phosphorylation state, which is regulated by mTOR (the mammalian/mechanistic Target of Rapamycin) kinase. Phosphorylated 4E-BPs dissociate from eIF4E and enable initiation of cap-dependent translation, whereas hypophosphorylated 4E-BPs act as inhibitors of translation [28]. In the brain, CYFIP1 (Cytoplasmic FMRP—Fragile X Mental Retardation Protein-Interacting Protein) acts as a 4E-BP [40]. CYFIP1 possesses an eIF4E-interacting domain, the 'reverse L shaped' structure, and thus, it is presumed to compete with 4E-BPs for the canonical eIF4E-binding motif [39,40]. The 'reverse L shaped' structure of CYFIP1 peptide includes two α-helical turns, stabilized by two internal salt bridges between residues Asp 742–Arg 744 and Glu 748–Lys 750 [40]. Importantly, these two salt bridges stabilize CYFIP1 in a favourable conformation to interact with Glu 132 of eIF4E through Lys 743 [40]. On the other hand, MNKs stimulate cap-dependent translation by phosphorylating eIF4E on Ser 209, upon activation by ERK (Extracellular signal-regulated kinase 1 and 2) or by the p38 pathway [41,42]. In particular, upon phosphorylation, MNKs interact with the scaffolding protein eIF4G, which acts as a docking site that brings MNKs and eIF4E into close proximity, thus facilitating eIF4E phosphorylation [43,44]. The role of eIF4E phosphorylation on Ser209 has been highly debatable. Several studies have linked eIF4E phosphorylation with cancer development [45–47] and the pathophysiology of neurodevelopmental and neuropsychiatric disorders [48,49]. Furthermore, phosphorylation of eIF4E seems to play a role in the export of mRNAs from the nucleus [50]. With regard to the mRNA-cap structure recognition, it has been shown that phosphorylated Ser 209 forms a salt-bridge clamp with Lys 159, which is located within the cap-binding site of eIF4E [51]. Interestingly, it is suggested that the salt-bridge clamp leads to a conformational change of the binding site that likely blocks the docking of the cap structure, resulting in an unfavourable energetic microenvironment, due to electrostatic repulsion between cap and phosphoserine [51]. Recently, in the brain and in the form of long-lasting synaptic plasticity, the long-term potentiation (LTPBrain-Derived Neurotrophic Factor) in the hippocampus, downstream of (BDNF), MNKs were shown to regulate early and late LTP, respectively, via CYFIP1/FMRP and 4E-BP2 repressor translation initiation complexes [52] Mammalian site 20-like kinase. Interestingly, eIF4E can be also phosphorylated on T55, by the (MST1), which was recently identified to inhibit the translation of a subset of mRNAs, but paradoxically bolster long non-coding RNA (lncRNA) translation [53]. T55 phosphorylation of eIF4E compromises its ability to bind the cap structure. This results in translation inhibition of the mRNAs encoding eIF2a and eEF2 (eukaryotic elongation factor 2), and CCT2 (chaperone protein T-complex protein 1 subunit β). On the contrary, it activates translation of lncRNA *linc00689* [53]. Linc00689 is an autism-associated primate-specific lncRNA, which is found upregulated in ASD (Autism Spectrum Disorder) cortex [54].

eIF4E is also regulated by eIF4E-transporter protein (4E-T) and eIF4E-homologous protein (4E-HP) proteins. 4E-T is present in processing bodies (P-bodies), binds to eIF4E and regulates the translation and decay of a subset of mRNAs [55]. In the developing cortex, an eIF4E1/4E-T-complex is present in granules and contains proneurogenic basic helix–loop–helix (bHLH)-containing mRNAs and thus its disruption leads to enhanced neurogenesis and depletion of neural precursors [55]. Work from the same group revealed a Smaug2 (Protein Smaug homologue 2) protein/Nanos1 (Nanos homologue 1) mRNA complex that is present in cytoplasmic granules together with 4E-T in neuronal precursors, and exerts translational control during brain development on transcriptionally primed cells, to dictate the balance between stem cell state and neurogenesis [56] Pumilio2. Furthermore, (Pum2) and 4E-T were shown to form repressive complexes, which regulate the translation of neuronal identity mRNAs in radial glial precursors of the mouse brain, thus regulating neuronal specification [57]. Another 4E-BP, 4E-HP, was first shown to participate in non-canonical translational control via tethering of mRNA 5′ and 3′ ends [58]. Subsequent work revealed that 4E-HP participates in translational control mechanisms linked to the Ribosome-Associated Quality Control of faulty mRNAs [59], RNA decay [60], antiviral immunity [61] and development [62]. 4E-HP was also linked to hypoxia-induced repression of protein synthesis. While most mRNAs are translationally repressed under hypoxia, a complex that includes the oxygen-regulated HIF-2α (hypoxia-inducible factor 2α), the RNA-binding protein RBM4 and 4E-HP promotes the translation of a subset of mRNAs by capturing their 5′ UTR [63], which is highly relevant for cancer cells [64].

Under certain conditions, such as cellular stress or viral infection, a cap-independent translation is used by eukaryotic organisms. This alternative translation mechanism bypasses the initiation scanning and the m7G-cap-recognition step (reviewed in [65]). Ribosomes are recruited in specific regions of the 5′ end of the mRNAs, the Internal Ribosomal Entry sites (IRES), originally discovered in viruses [66,67]. Notwithstanding the fact that cellular mRNA IRES activity observed in plasmid-based assays is usually weak, this mechanism is also proposed to exist for eukaryotic mRNAs [68]. IRES elements identified in cellular mRNAs are difficult to classify, as they are all different from one another in sequence and lack a unifying structural motif, which suggests that they may constitute a plastic/flexible cellular response akin to intrinsically disordered proteins [69]. On the other hand, several reports have proved the activation of IRES during physiological processes, such as mitosis, cell differentiation, neuron plasticity and apoptosis, revealing a plethora of transcription factors, transporter proteins, receptors and growth factors, encoded by IRES-containing transcripts [70]. Overall, direct evidence for the role of cellular mRNA IRES *in vivo* is controversial [71].

Although elongation is a highly regulated process, that consumes almost all the energy and the nutrients required during translation, such as glucose, ATP (Adenosine 5′-triphosphate) and amino acids, it is a much less studied process [47]. A complete cycle of elongation consists of two stages (figure 2b). During the first stage, an amino acid-transfer RNA (aa-tRNA) is recruited to the ribosome A site, by the GTP-bound eukaryotic elongation factor 1A (eEF-1), followed by tRNA and cognate codon pairing, which leads to GTP hydrolysis by eEF1A [72]. Simultaneously, the ribosome changes its conformation, stimulating the shift in position of the aa-tRNA from A to P site, and the tRNA carrying the polypeptide chain, from P to exit (E) site. This sequence of events catalyses the peptide bond formation and the transfer of the polypeptide chain to the aa-tRNA, leading eventually to the extension of the nascent polypeptide [73]. During the second stage of elongation, the eEF2 enters the A site of the ribosome and stimulates a change in ribosomal conformation. Finally, the ribosome translocates and a new cycle of elongation can start [72].

Translation is also regulated at the elongation phase. Several factors have been identified to affect the rate of elongation, particularly impacting the codon decoding step or the ribosome translocation [27]. Codon decoding is a tRNA-dependent process and can therefore be affected by the relative abundance of each tRNA, the cognate to near-cognate tRNA ratio, or the tRNA aminoacylation process. A cognate aa-tRNA can pair with the first two bases of the mRNA codon, following Watson–Crick interactions or non-Watson–Crick interactions at the third base, the so-called wobble position. A tRNA that does not meet the above criteria is designated near-cognate tRNA [74]. It has been shown that tRNAs are differentially expressed among tissues and even among different cell types. For example, cancer cells can reprogramme tRNA expression and change the abundance of specific tRNAs required for the expression of cancer-related genes [75,76]. On the other hand, some codons can be decoded by near-cognate tRNAs, through a mismatched nucleotide. In some cases, they show decreased decoding speed, compared to their synonymous codons [77]. Aminoacylation is the process of tRNA pairing with the corresponding amino acid, by a cognate aminoacyl-tRNA synthetase (ARS), and it is, therefore, very important for tRNA functionality and proper mRNA translation in cells, as it controls the distribution of charged tRNAs in the cell [78]. Additional control of translation elongation is provided by the phosphorylation of eEF2 on threonine-56, by eEF2 kinase (eEF2). Phosphorylation

leads to the inhibition of eEF2 activity to promote ribosome conformation, by physically blocking its entry to A site, and thus limiting elongation [47].

# 3. Translational control in cells of the vascular system

Several signalling pathways are activated during vasculogenesis and angiogenesis, providing the appropriate positive and negative regulators for vessel formation. The pivotal role of the vascular system in supplying the mammalian cells with oxygen and nutrients, and therefore supporting their normal development, presupposes the existence of effective and rapid-response mechanisms, for ECs, VSMCs to adapt to external/environmental signals, such as mechanical force or in pathologic conditions, including wound healing and oncogenesis in cancer. Translational control is a mechanism that allows for rapid changes of ECs and VSMCs function and phenotype.

The majority of studies investigating translational control in ECs and VSMCs were performed in the cardiovascular system. The heart is considered to be the organ that is impacted the most by biomechanical forces [79]. To maintain its function under biomechanical stress conditions, heart cells (cardiomyocytes; CM) undergo physiological hypertrophy [80,81]. It is known that mechanical forces activate the mTOR signalling pathway and regulate protein synthesis in heart (CM) and vascular cells (ECs and VSMCs) [79]. At baseline, VSMCs are shielded from mechanical stress, while ECs are more responsive to mechanical forces, such as those leading to mTOR activation. Inappropriate EC activation due to mechanical force or other factors may lead to VSMCs activation [82,83]. Alteration of mechanical forces applied on VSMCs induces changes in protein synthesis and expression of pro-inflammatory molecules, in lungs [84], but also endoplasmic reticulum (ER) stress [85], while cyclic strain activates mTOR signalling in VSMCs [86]. VSMCs have been linked to mTORC1/mTORC2 (mTOR complex 1/mTOR complex 2) in models of pulmonary hypertension. Rapamycin, an inhibitor of mTORC1, decreased VSMCs proliferation and vessel remodelling in hypertensive rats [87]. Genetic deletion of Raptor, which led to mTORC1 disruption, improved VSMC proliferation in a pulmonary hypertension mouse model [88]. Conversely, mTORC2 disruption, via Rictor deletion, in mice, led to spontaneous pulmonary hypertension [88]. Another link of VSMCs to the translation machinery is the observation that RPL17 (large ribosomal protein L17) acts as a VSMC growth inhibitor [89].

In ECs, fluid shear stress activates the mTOR pathway, through phosphorylation of the mTOR target p70S6 K (ribosomal protein S6 kinase β-1), and facilitates the translation initiation of specific mRNAs, such as the proto-oncogene Bcl-3 factor mRNA [90]. In particular, it has been shown that Bcl-3 expression decreases in response to translation inhibition by rapamycin, but remains unaltered after transcription inhibition [90]. A link between mTOR signalling and atherosclerosis development was recently identified [91]. Following low shear stress, mTOR is activated, as evidenced by increased 4EBP1 phosphorylation, while the AMPKα (AMP-Adenosine monophosphate-activated protein kinase α) signalling pathway is inhibited. These two events, together with a concomitant blockade of the autophagosomal and lysosomal fusion, impair autophagy and lead to endothelial cell apoptosis, which eventually facilitates atherosclerosis development [91]. Interestingly, it was shown that, during the recovery of heat-denatured ECs or denatured dermis in rats, autophagy was upregulated and promoted angiogenesis, by an AMPK/Akt/mTOR-mediated mechanism [92]. Akt or Protein kinase B is a serine/threonine-specific protein kinase, which is part of the PI3 K (Phosphoinositide 3-kinase)/ Akt/mTOR signalling pathway. Phosphorylated Akt activates mTORC1 by direct phosphorylation of the PRAS40 (proline-rich Akt substrate of 40 kDa) and the TSC2 (tuberous sclerosis protein2) [93]. In turn, mTOR phosphorylates Akt, leading to its full activation [94].

The PI3 K/Akt pathway has a pivotal role in many cellular processes. Several lines of evidence support a significant role of PI3 K/Akt pathway in regulation of angiogenesis in both normal tissue and in cancer (reviewed in [95]). The PI3 K/Akt pathway can be activated by the RAS GTPases [96], by increased growth factor expression [97], or after phosphatase and tensin homologue (PTEN) deactivation [98,99]. Furthermore, the activity of the angiogenic VEGF is partially mediated by the PI3 K pathway [95]. However, the core mechanism of increased VEGF secretion is hypoxia, which is observed in tumour cells.

During hypoxia, the transcription factor hypoxia-inducible factor-1 (HIF-1) is stabilized, dimerizes and induces the transcription of a plethora of target genes, including VEGF [100–102]. Interestingly, activation of the PI3 K/mTOR pathway increases HIF-1a protein levels by increasing HIF-1a translation, leading to upregulation of VEGF expression and thus promoting angiogenesis [103,104]. Surprisingly, in a transgenic rodent model of conditional HIF-1 induction, it was shown that in the

absence of VEGF, ECs can produce sprouts, but are unable to form new vessels, suggesting a vital role of VEGF in neovascularization [105]. More recently, the effects of mTORC1 and mTORC2 inhibition on angiogenesis were identified in ECs. Sustained mTORC1 inhibition led to increased Akt1 activation and also, pre-sensitized ECs to angiogenic cues, whereas specific mTORC2 inhibition resulted in a dramatic reduction of angiogenic sprouting and a significant decrease in the length of the sprout extension, in VEGF-stimulated angiogenesis [106]. Further support on mTORC2's role in angiogenesis was shown with conditional loss of function in *in vitro* and *in vivo* experiments, targeting mTORC1 and mTORC2. In particular, EC ablation of Rictor, but not Raptor, inhibited VEGF-induced EC proliferation, vascular assembly and angiogenesis [107]. mTORC2 seems to regulate ECs proliferation and angiogenesis through two downstream effectors, Akt and PKCa (Protein Kinase C α), indicated by the decreased phosphorylation of both molecules, after Rictor deletion [107]. Interestingly, it was recently shown that VEGF activates Unfolded Protein Response (UPR) mediators, such as Activating Transcription Factor 6 (ATF6) and PKR - Protein kinase R-like ER kinase (PERK), dependent on Phosphoinositide phospholipase C $\gamma$ (PLC$\gamma$)-mTORC1 crosstalk, without accumulation of unfolded proteins in the ER. This reveals that VEGF signals to the ER and UPR, suggesting further links to protein synthesis [108]. Taken together, these findings in conjunction with the aberrant activation of ER stress in VSMCs following mechanical stress [85] suggest that pathways such as the Integrated Stress Response (ISR) and their link to translational control may constitute a convergence mechanism for vascular cell biology. The ISR was previously shown to be pivotal in brain health and disease [109,110] and to regulate synaptic plasticity, learning and memory [111].

# 4. Translational control of VEGF

VEGF is one of the key regulatory molecules of angiogenesis. Its expression is regulated at multiple levels, post-transcriptionally. Several mechanisms of VEGF mRNA-specific translation have been identified (extensively reviewed in [112]). The 5′ UTR region of VEGF-A mRNA contains three in-frame alternative CUG start codons and two IRESs (IRES-A, IRES-B), which allow for alternative translation initiation [114–115]. IRES translation of the VEGF mRNA is induced under local environmental stress conditions, such as hypoxia [116]. Two IRES *trans*-acting factors controlling translation of VEGF mRNA have been identified; the positive regulator kinase Mitogen-activated protein kinase 3 (MAPK3) [117] and the DEAD-box RNA helicase 6 (DDX6), which inhibits the VEGF IRES-mediated translation under normoxic conditions [118]. Furthermore, a short uORF, 186 nucleotides upstream of the main ORF, has been identified in VEGF 5′UTR [119]. Interestingly, this unique uORF is located within one of the IRESs of VEGF 5′UTR and, unlike the hindering role of uORFs to the scanning ribosomes, it differentially regulates the expression of the different VEGF isoforms [119]. A significant number of microRNAs (mRNAs) have been reported to bind within the 3′UTR region of VEGF, and inhibit its expression, thus providing an additional regulatory mechanism [112]. It has been shown that miR-16 inhibits IRES-B translation of VEGF but does not affect IRES-A translation, revealing for the first time an isoform-specific miRNA-mediated inhibition of translation [120]. A novel mechanism of VEGF translation regulation has been identified which preserves basal levels of expression, under translation repression [121]. This translational trickle mechanism was identified in myeloid cells and protects a small amount of VEGF mRNA from GAIT ($\gamma$ interferon inhibitor of translation element) complex-mediated translational inhibition [121]. The GAIT complex consists of the ribosomal protein L13a, glyceraldehydes-3-phosphate dehydrogenase (GAPDH), NS1-associated protein 1 (NSAP1) and glutamyl-prolyl tRNA synthetase (EPRS) [122]. A C-terminus truncated EPRS[N1] protein was discovered, which binds and protects a small amount of VEGF mRNA GAIT target mRNAs, providing low levels of VEGF expression [121]. Recently, the AU binding factor 1 (AUF1) was shown to have a dual control on VEGF mRNA expression, and, therefore, on angiogenesis [123]. AUF1, also known as heterogeneous nuclear ribonucleoprotein D (hnRNP D), was the first AU-rich element-binding protein (ARE-BP) identified to act as an mRNA destabilizer [124]. However, its function has been proven to be more complex, including mRNA stabilization [124], splicing [125] and translational repression [126]. AUF1 stabilizes both the VEGF-A and the HIF-1$\alpha$ mRNAs, thus facilitating angiogenesis [123].

# 5. Local translation

Localized translation is an alternative mechanism of gene expression regulation that offers extreme precision and spatio-temporal control of protein synthesis in compartmentalized cells [22]. Proteins are

expressed in the cellular compartment where they are needed. Local translation has been proven to be a highly conserved process and holds a key role in supplying the cells with new proteins, on-site and at short time scales, bypassing the delay in protein transport. It is widely observed in migrating cells, such as fibroblasts, myoblasts and neurons [127], and is crucial for cell development and survival. Cell migration consists of three smoothly coordinated steps: protrusion, contraction and retraction [128]. Protrusion and focal adhesions are formed at the leading edge of the migrating cell, followed by cell polarization. Then the nucleus moves and finally, the cell body translocates. The successful completion of this cycle is crucial for normal cellular functions (e.g. development and wound healing) but also in pathologic conditions such as tumour metastasis, and it is regulated by the local expression of specific factors [127]. Two very well-studied molecules of local translation in migrating cells are β-actin and the regulator of the actin cytoskeleton, Actin Related Protein 2/3 (Arp2/3) complex, both found in the protrusions of the migrating cells (extensively reviewed in [127]).

The following criteria have been proposed for the establishment of protein synthesis at a local level: (i) the localization of mRNA, ribosomes and translation regulatory elements at specific subcellular compartments, (ii) the detection of nascent proteins and (iii) a decrease in protein levels after blockade of local synthesis [129]. A very common localization mechanism of mRNAs that encode cytoplasmic proteins is based on specific RNA sequences, the zipcodes [130]. A zipcode can be found in the 3′ or 5′ UTR region of the mRNA, and holds the information for the mRNA translocation from the nucleus to a specific compartment, which is mediated by RNA-binding proteins (RBPs), the TAFs (*trans*-acting factors) [127]. External signals, like trophic factors, neurotransmitters and guidance cues, shape the local translatome and determine the response of the cell. Transported mRNAs and RBPs form the mRNP (messenger ribonucleoprotein) granules, which also contain regulatory mRNAs and, together with motor proteins, define the subcellular compartment a specific mRNA will be transferred for translation [131]. Several mRNPs have been identified, including the P-bodies, the stress granules, and the neuronal RNA granules, which could be the same mRNP complex at different phases of its life cycle [22]. It was initially suggested that related mRNAs are transported and stored together at the same mRNP. For example, Calcium/calmodulin-dependent protein kinase type II (CaMKII), neurogranin and activity-dependent cytoskeleton-associated protein (Arc) mRNAs, all encoding synaptic plasticity-related proteins, were found to co-localize to the same mRNP granule in neuronal dendrites of hippocampal neurons [132]. However, recent studies have revealed that each granule can contain one or two different mRNA transcripts, and surprisingly, there is evidence showing that mRNAs or mRNPs are regulated via a neuronal activity-dependent mechanism, in neuronal dendrites [134–135].

Local translation has been extensively studied in neurons. Being migrating cells, neurons use their growth cone as a detector of the local environmental cues to navigate their growing axons, and after reaching their destination, they finally branch and form synapses in the dendrites and the cell soma. Synapses are dynamic structures, capable of changing their strength and efficacy, a process mediated by neuronal activity, known as synaptic plasticity. Long-lasting forms of synaptic plasticity, like long-term memory formation, require new mRNA and/or protein synthesis, in specific neuronal compartments, e.g. dendrites [137–138]. Interestingly, long-term plasticity can occur at some but not all synapses of a single neuron, suggesting that there are specific mechanisms that compartmentalize the modifications of the proteome to those synapses undergoing plasticity [129]. The first evidence was shown in hippocampal pyramidal neurons, severed from their soma, where BDNF-induced LTP was blocked by protein synthesis inhibitors, providing direct evidence of a dendritic translation-dependent synaptic plasticity [139]. Similarly, other studies have shown that the late phase of LTP (L-LTP) and the metabotropic-Glutamate receptor (mGluR) dependent long-term depression are regulated by localized translation in dendritic neuronal compartments [138,140,141]. Several additional studies in *Aplysia* have revealed that translation-dependent forms of memory, including long-term and intermediate forms of memory, are regulated locally and are often independent of somatic translation [142,143]. In addition, local protein synthesis is required for local homeostatic scaling, a type of homeostatic plasticity [144] shown in the dendrites of rat hippocampal neuron cultures, after local perfusion of a protein synthesis inhibitor [143]. Apart from synaptic plasticity, local translation is important for axonal development [145,146]. β-Actin needs to be asymmetrically translated, in response to the environmental cues that guide the AGC. Furthermore, in adult axons, local translation is required for initiation of injury response through axonal translation of importin β1 [147], but also for the axon survival and maintenance [148,149]. In a recent study, conducted in rodents, it was shown that not only post-synaptic compartments but also pre-synaptic terminals from hippocampus and cortex are translationally competent [150], as revealed by the translational machinery and the abundant protein synthesis observed in the pre-synaptic terminals of the mouse brain [150].

Despite the fact that neurons are central in the field of local translation, naturally other cell types of the NVU may employ compartmentalized translation to perform various functions. ECs respond to extracellular stimuli via specific protein polarity and localization [151]. RNA sequencing of protrusions of migrating primary human umbilical vein ECs (HUVECs) and comparison with other cell types, among which neurons revealed a cluster of 5 mRNAs, which exhibited universal targeting to protrusions in all cell types tested and harboured a distinct 3′UTR sequence responsible for targeting [152]. Among the mRNAs in this cluster, polarization of Ras-related protein Rab-13 (RAB13) was shown to be required for blood vessel morphogenesis. Furthermore, local translation of polarized RAB13 generated a pool of localized nascent RAB13, which may confer distinct function roles associated with membrane remodelling [152]. Interestingly, Rab13 was also shown to regulate neurite outgrowth [153]. Local translation also occurs in perivascular astrocyte processes (PVAPs) and perisynaptic astrocytic processes (PAPs) in astrocytes for a specific subset of mRNAs, and is related to memory [155–156].

# 6. Translational control of neurovascular development and neurodevelopmental disorders

Neurodevelopmental disorders (NDDs) are a group of conditions affecting physical, learning, language or behaviour areas. NDDs are among the most common chronic disorders in children worldwide, affecting approximately 1% of the world population (Centres for Disease Control and Prevention, USA, and [157]). These conditions emerge during the developmental period, in most cases persist to adulthood and may last throughout a person's lifespan. NDDs are usually accompanied by severe comorbid conditions among which are epilepsy, depression, anxiety, sleep disorders, metabolic disorders and neuroinflammation [158]. According to the Diagnostic and Statistical Manual of Mental Disorders, Fifth Edition (DSM-V) [159], NDDs include intellectual disability/developmental disorder (ID/DD), communication disorders, ASD, schizophrenia (SZ), attention deficit hyperactivity disorder (ADHD), specific learning disorders and motor disorders. Neurovascular systems of the brain are not traditionally associated with NDDs, but rather with neurodegenerative disorders. In Alzheimer's disease, BBB breakdown and altered blood flow are causally linked to vascular endothelium dysfunction [160,161]. Similarly, studies in Parkinson's disease (PD) have shown disruptions in BBB, which induced neuroinflammation and accumulation of toxic forms of α-synuclein [162]. Despite the fact that there are no reports of elevated α-synuclein in ECs, post-mortem sections from PD patients brain have revealed perivascular accumulation of α-synuclein, suggesting an α-synuclein-induced EC activation, which could lead to abnormal elevation of oxidative stress, and eventually promote neuronal loss [163].

With regard to NDDs, the vascular hypothesis of SZ proposes a vascular component for the pathophysiology of SZ, according to which damage in the microvascular system disturbs the normal oxygen and energy supply of the brain, contributing to the impaired brain function [164]. There are a substantial number of human and experimental studies identifying possible mechanisms that induce increased oxidative stress and neuroinflammation, such as the increased VEGF activity [165,166] and microglial activation [168–169]. These mechanisms (extensively reviewed in [170]) strongly suggest NVU dysfunction and BBB hyperpermeability in SZ brain, providing a link between the vascular and the nervous system in the disease; however, whether the vasculature deficits are a cause or a consequence of the deficient brain development still remains unclear [171]. Recent studies in ASD patients revealed alterations in angiogenesis in post-mortem brain, shown by aberrant staining of pericytes [172] and changes in cerebral blood flow [173]. Furthermore, developmental ASD-linked 16p11.2 haploinsufficiency in ECs results in neurovascular and behavioural changes in adult mice and in defective angiogenesis in induced-pluripotent-stem-cells (iiPSCs), derived from human carriers of the 16p11.2 deletion [174]. Yet, the precise contribution of neurovascular deficits to the pathophysiology of neurodevelopmental disorders, such as ASD, remains elusive.

ECs-specific conditional mouse models of the Gamma aminobutyric acid (GABA) pathway (Gabrb3ΔTie2-Cre and VgatΔTie2-Cre) revealed that partial or complete loss of GABA in ECs during embryogenesis impairs cortical interneuron migration, leading to behavioural deficits in adult mice [175]. Moreover, activation of GABA signalling in forebrain ECs promoted their migration, angiogenesis and acquisition of BBB properties [175]. These results, in conjunction with the fact that inhibitory but not excitatory cortical neurons require vascular support for proper neurogenesis [10], highlights a significant role for the neurovascular interplay during cortical development, with wider implications for the cortical E/I balance. E/I imbalances have been linked to NDDs, such as ASD [176]. On the other hand, decreased VEGF has been associated with NDDs including schizophrenia,

**Table 1.** Translational control in neurodevelopmental disorders.

| neurodevelopmental disorder | genes/pathways affected | translational dysregulation | reference |
|---|---|---|---|
| ASD | SNPs in CYFIP1, CYFIP1-FMR1-eIF4E | increased CYFIP1 mRNA, translation repression | [174,181,182] |
| | FMRP | increased mTOR, ERK and p70S6 K activity | |
| | Akt/mTOR | decreased TSC2, PTEN and GSK3 | |
| | 16p11.2 microdeletion | | |
| fragile X mental retardation (FXS) | FMR1 mutations | increased mTOR phosphorylation/ activity, elevated pAKt | [183,184,185] |
| | PI3 K/mTOR/p70S6 K | increased p70S6K1/pS6 | |
| | p70SK1 | increased expression of CYFIP2 | |
| | | elevated phosphorylation in S6, eIF4B, mTOR, ERK | |
| tuberous sclerosis | TSC1 and TSC2 mutations | constitutive mTOR signalling | [186,187] |
| depression/depression-like behaviour | ERK/p38-MNK1/2–eIF4E | reduced ERK1/2 mRNA, protein, activity, decreased eIF4E phosphorylation | [188,189] [190,191,192,193] |
| | mTOR/p70S6 K/eIF4B pathway | reduced mTOR activity, reduced phosphorylated eIF4B | |
| | | decreased phosprylated4E-BP1, ERK and Akt | |
| schizophrenia | SNPs in CYFIP1 | increased pAkt, pS6 | [194,195,196] |
| | DISC1, Akt–mTOR | | |

bipolar disorder and autism [170,177]. Moreover, both perinatal and intrauterine hypoxic insults are highly associated with NDDs [178]. VEGF transcription is upregulated by HIF1 in hypoxia [101]. 4E-HP knock-down downregulated VEGF protein levels and secretion, both in normoxia and hypoxia. Inhibiting 4E-HP also led to lower tube formation and endothelial cell migration and coincidentally 4E-HP and VEGF protein expression emerge at embryonic day 10.5. Both 4E-HP and VEGF were previously linked to neurogenesis, learning and memory [101,179,180]. This raises the exciting prospect that the 4E-HP-VEGF interplay may be relevant to vascular and potentially neurovascular co-development. To our knowledge, there are currently no studies directly associating mRNA translation control mechanisms with neurovascular development in NDDs, although there is a growing list of publications supporting a strong link between NDDs and vascular deficits.

Dysregulated translational control in NDDs has gained a lot of attention as a risk factor in these conditions (table 1). The recent concept of 'mTORopathies' describes the categorization of a large class of NDDs with impaired mTORC function [197]. Upregulated mTOR signalling has been linked to ASD [199–200], while downregulated in major depressive disorder (MDD) [192,193,201,202] (table 1). Additionally, mutations in genes encoding core factors of mRNA translation have been revealed. For example, the RNA-binding protein FMRP, encoded by the FMR1 gene, inhibits translation initiation and mutations in the gene causes the Fragile X Syndrome (FXS), the most common inherited cause of ASD (table 1). CYFIP1 mediates the activity of FMRP on translation repression. Mutations in CYFIP1 have been linked to both ASD [182] and SZ [196] (table 1).

In conclusion, neurons and vascular cells co-develop in the developing mammalian brain primarily under the control of guidance cues. However, regulation of gene expression at the level of translational control in neurovascular cells emerges as a novel avenue that may also be coordinated in these two key systems of the brain, with wider implications for NDDs.

Data accessibility. This article has no additional data.

Authors' contributions. All authors gave final approval for publication. K.C. and C.G.G. conceived the topic of this review. K.C. and E.S. carried out literature review and wrote/edited parts of the manuscript. M.M., S.B., E.B. and T.F. wrote/ edited parts of the manuscript. K.C., C.M. and C.G.G. wrote/edited and performed final editing of the manuscript.

Competing interests. We declare we have no competing interests.

Funding. C.G.G. was supported by a Fondation Santé grant. C.G.G. and C.M. are recipients of a FORTH Synergy grant.

Acknowledgements. The authors sincerely apologize to all colleagues whose work has been omitted due to space and scientific topic limitations.

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
