## [Peer Review File · Royal Society Open Science]

Review History

RSOS-211088.R0 (Original submission)

Review form: Reviewer 1

Is the manuscript scientifically sound in its present form?

Yes

Are the interpretations and conclusions justified by the results?

Yes

Is the language acceptable?

Yes

Do you have any ethical concerns with this paper?

No

Have you any concerns about statistical analyses in this paper?

No

Recommendation?

Accept with minor revision (please list in comments)

Comments to the Author(s)

See attached file (Appendix A).

Review form: Reviewer 2 (Julia Clarke)

Is the manuscript scientifically sound in its present form?

Yes

Are the interpretations and conclusions justified by the results?

Yes

Is the language acceptable?

Yes

Do you have any ethical concerns with this paper?

No

Have you any concerns about statistical analyses in this paper?

No

Recommendation?

Accept with minor revision (please list in comments)

Comments to the Author(s)

The manuscript by Chalkiadaki and colleagues entitled "Translational control in neurovascular brain development" is an interesting and extensive review of the translational control mechanisms taking place in brain and endothelial cells during developmental stages. The manuscript is clear and well written, and is clearly an important contribution to the field.

My suggestions to improve the article are:

- the discussion on translational control of neurovascular development in neurodevelopmental disorders is shallow and focuses mostly on autism. This section could be improved by including data on schizophrenia, early life infections with long-lasting consequences, and others. In addition, the manuscript would improve if a final figure summarizing the main events in translational control during health and neurodevelopmental diseases was added.

-I suggest authors include a final section of Discussion and perspectives in the field, especially concerning translational control of neurovasculature in neurodevelopmental diseases, as this seems to be a fertile research ground.

Decision letter (RSOS-211088.R0)

Dear Dr Gkogkas

On behalf of the Editors, we are pleased to inform you that your Manuscript RSOS-211088 "Translational control in neurovascular brain development." has been accepted for publication in Royal Society Open Science subject to minor revision in accordance with the referees' reports. Please find the referees' comments along with any feedback from the Editors below my signature.

Please submit your revised manuscript and required files (see below) no later than 7 days from today's (ie 09-Aug-2021) date. Note: the ScholarOne system will 'lock' if submission of the revision is attempted 7 or more days after the deadline. If you do not think you will be able to meet this deadline please contact the editorial office immediately.

on behalf of Dr Robson da Costa (Associate Editor) and Catrin Pritchard (Subject Editor)
openscience@royalsociety.org

Associate Editor Comments to Author (Dr Robson da Costa):

Comments to the Author:

This is an important and well-written review. However, there is a need to improve the quality of the text.

Authors should take into account all points raised by reviewers in order to improve the writing of the manuscript.

Reviewer comments to Author:

Reviewer: 1

Comments to the Author(s)

See attached file (Chalkiadaki et al review.pdf)

Reviewer: 2

Comments to the Author(s)

The manuscript by Chalkiadaki and colleagues entitled "Translational control in neurovascular brain development" is an interesting and extensive review of the translational control mechanisms taking place in brain and endothelial cells during developmental stages. The manuscript is clear and well written, and is clearly an important contribution to the field.

My suggestions to improve the article are:

- the discussion on translational control of neurovascular development in neurodevelopmental disorders is shallow and focuses mostly on autism. This section could be improved by including data on schizophrenia, early life infections with long-lasting consequences, and others. In addition, the manuscript would improve if a final figure summarizing the main events in translational control during health and neurodevelopmental diseases was added.

-I suggest authors include a final section of Discussion and perspectives in the field, especially concerning translational control of neurovasculature in neurodevelopmental diseases, as this seems to be a fertile research ground.

===PREPARING YOUR MANUSCRIPT===

===PREPARING YOUR REVISION IN SCHOLARONE===

Author's Response to Decision Letter for (RSOS-211088.R0)

See Appendix B.

Decision letter (RSOS-211088.R1)

Dear Dr Gkogkas,

I am pleased to inform you that your manuscript entitled "Translational control in neurovascular brain development." is now accepted for publication in Royal Society Open Science.

on behalf of Dr Robson da Costa (Associate Editor) and Catrin Pritchard (Subject Editor)
openscience@royalsociety.org

Appendix A

This review addresses the translational control of gene expression in the nervous system and relates this to the interaction between neuronal cells and vascular cells required for brain development. Topics covered are appropriately detailed, informative and relevant.

The impact of the review can be enhanced by linking the different sections better. At the moment, the text reads like a combination of sections written by the various authors. Making better links between different sections and reminding the reader of the relevance of examples discussed will enhance the impact of this review.

Major issues:

Make better use of figures and figure legends. In Figure 1, expand explanation of C and D. C - what are red/blue dots? D - spell out what the connections are between components of neurovascular unit. In Figure 2, panel B implies incorrect tRNA-anticodon base pairing- this should be amended. Figure 2 legend should be more extensive with better explanations of factors and steps.

The section on IRES (232 onwards) is ambivalent about the physiological function of IRES in translation of cellular mRNAs, while in the section describing VEGF translational control (367..) there is an example of IRES controlled translation initiation. The authors should reconcile the message of these two sections.

Minor issues:

Use of brackets – should be left for definition of abbreviations while explanations put into brackets should be incorporated into the sentence.

Be consistent in the definition of abbreviations. Currently, in some instances the abbreviation is given in brackets (e.g. 430), and in others the full name (e.g. 307).

Specific comments:

- 59 full stop missing
- 78 rely on mRNA transport (no need to define messenger RNA)
- 88 full stop in wrong place
- 110 delete "refs"
- 148 "at" not "in" the 5'-end
- 157 what do you mean with "nutrients" used during translation?
- 160-170 consider combining this section with the one starting at 246 where you go into more detail on translation elongation
- 174 define which "complex" you mean
- 177 typo "phopso"
- 185 explain how MNKs regulate eIF4E activity
- 195 explain the significance of "reverse L-shaped structure"?
- 197 Consider combining section on MNK with the one earlier on this page.
- 198 Discuss consequences of S209 phosphorylation.
- 203 Can you include examples of targets by this regulation?
- 211 wrong bracket
- 249 explain cognate /near cognate tRNAs
- 253 split sentence starting "On the other hand.." into 2.
- 280 Change "EC, VSMC" to "ECs and VMSCs" here and elsewhere.
- 295 (in lungs) - remove brackets. Here and elsewhere use brackets for definition of abbreviations while explanations put into brackets should be incorporated into the sentence.
- 297 Explain the nature of the link between VSMCs and mTOR complexes.
- 298 add comma after mTORC1
- 326 consider replacing "several" in "pivotal role in several cellular processes" with "many"
- 330 check citation/bibliography

370 delete "ref"
395 in "Recently, the AUF1", delete "the"
421 delete "ref"
428 Correct citation Anon 2003
432 In sentence starting with "Transported mRNAs.." Explain what you mean with "regulatory mRNAs" - are these mRNAs encoding regulatory proteins? or mRNAs that have a regulatory function in mRNA form?
436 use the term "mRNP granules" instead of mRNPs here and elsewhere where talking about granules
445 "neuronal activity-dependent mechanism" this is vague and needs expanding – or does it relate to your next section? Then this should be made clear.expand on this - or make clear this is related to the next section
466 "often," remove comma
475 "recent study," remove comma
501 delete "ref"
522 Is it necessary to refer to the mouse genotypes? The findings are clear without them.
529 Expand on the significance of the E/I balance
539 Explain why this prospect is exciting

430 Explain what you mean with "regulatory mRNAs" - are these mRNAs encoding regulatory proteins? or mRNAs that have a regulatory function in mRNA form?
179 post-transcriptional, not post-translational
179 awkward sentence – consider rewording ("The 3' end of mature mRNAs is a poly(A) tail added in a post-transcriptional modification step..")
579/580 References in confused format – is "ANON" a name or does it stand for anonymous?

Appendix B

Point-by-point response to reviewers' comments

We would like to thank both reviewers for their constructive criticism. Please find below a point-by point response to their comments.

Reviewer #1

Major issues:

1a. Make better use of figures and figure legends. In Figure 1, expand explanation of C and D. C - what are red/blue dots? D - spell out what the connections are between components of neurovascular unit.

We have expanded the explanation of Figure 1C (Figure) and 1D (Figure legend, lines 118-120)

1b. In Figure 2, panel B implies incorrect tRNA-anticodon base pairing- this should be amended.

We have now expanded description of elongation in the figure legend so that the reader can correctly understand the biophysical events involving tRNA and the ribosome during translation elongation.

1c. Figure 2 legend should be more extensive with better explanations of factors and steps.

We have changed the Figure 2 legend into a more detailed format (lines304-318)

2. The section on IRES (232 onwards – now 242) is ambivalent about the physiological function of IRES in translation of cellular mRNAs, while in the section describing VEGF translational control (367 (now 388)) there is an example of IRES controlled translation initiation. The authors should reconcile the message of these two sections

We thank the reviewer for pointing this out. To address this comment we amended the text (lines 258-262)

Minor issues:

1. Use of brackets – should be left for definition of abbreviations while explanations put into brackets should be incorporated into the sentence.

We have corrected the usage of brackets wherever was needed (lines 44-45, 56-57, 58, 74, 106, 114, 134, 146, 155-156, 169, 211, 340, 342, 347, 349, 498, 500, 532, 578)

2. Be consistent in the definition of abbreviations. Currently, in some instances the abbreviation is given in brackets (e.g. 430), and in others the full name (e.g. 307).

We have now unified the use of brackets for abbreviations throughout the text.

Specific comments:

1. 59 full stop missing

Full stop has been added (line 60)

2. 78 rely on mRNA transport (no need to define messenger RNA)

Definition of mRNA has been removed (line 79)

3. 88 full stop in wrong place

Mistake has been corrected (line 89)

4. 110 delete “refs”

The word ‘refs’ has been removed (line 112)

5. 148 “at” not “in” the 5’-end

The word ‘in’ has been replaced with ‘at’ (line 152)

6. 157 what do you mean with “nutrients” used during translation?

We understand the reviewer’s comment and to address this we have modified the sentence and replaced the word ‘nutrients’ with ‘glucose, ATP and amino acids’, which are more specific (lines 266-267).

7. 160-170 consider combining this section with the one starting at 246 where you go into more detail on translation elongation

We agree with the reviewer and we have now combined the two sections (lines 265-278)

8. 174 define which “complex”

We have defined the reference to the word ‘complex’ Line 165

9. 177 typo “phopso”

Typo has been corrected (line 169)

10. 185 explain how MNKs regulate eIF4E activity

To address this comment, we have added a short description of the eIF4E regulation by MNKs. This is now included in lines 195-198.

11. 195 explain the significance of “reverse L-shaped structure”?

To address this comment, we have added an explanation of the “reverse L shaped structure” in the text (lines 189-193)

12. 197 Consider combining section on MNK with the one earlier on this page.

We thank the reviewer for this comment; however we have devoted this section on Mnks to highlight their important role in translational control in brain.

13. 198 Discuss consequences of S209 phosphorylation.

To address this comment, we amended the text to reflect this comment (lines 198-210)

14. 203 Can you include examples of targets by this regulation?

We have addressed this comment by adding the identified targets in the text (lines 217-222)

15. 211 wrong bracket

Typo has been corrected (line 228)

16. 249 explain cognate /near cognate tRNAs

We have addressed this comment by specifying the difference between cognate/near cognate tRNAs (lines 283-286)

17. 253 split sentence starting “On the other hand..” into 2.

We have split the sentence (lines 290-292)

18. 280 Change “EC, VSMC” to “ECs and VMSCs” here and elsewhere

We have now replaced “EC, VSMC” to “ECs and VSMCs” in the text

19. 295 (in lungs) - remove brackets. Here and elsewhere use brackets for definition of abbreviations while explanations put into brackets should be incorporated into the sentence.

We have removed brackets here (line 342) and elsewhere and we have incorporated explanations into the sentences (see also minor comment #1)

20. 297 Explain the nature of the link between VSMCs and mTOR complexes.

We are unsure what we would need to address for this comment, as following sentence 344, we have explained the link between VSMVs and mTOR in the closing 2 sentences of this sections.

21. 298 add comma after mTORC1

We have added a comma after “mTORC1” (line 345)

22. 326 consider replacing “several” in “pivotal role in several cellular processes” with “many”

We have now replaced “several” with “many” (line 373)

23. 330 check citation/bibliography

Citation/bibliography has been checked and corrected (line 377).

Reviewer #2

1a) the discussion on translational control of neurovascular development in neurodevelopmental disorders is shallow and focuses mostly on autism. This section could be improved by including data on schizophrenia, early life infections with long-lasting consequences, and others.

To address this comment we have amended the text, by extending the discussion on the neurovascular deficits observed in schizophrenia. However, current literature on translational control of neurovascular development in NDDs is very limited (lines, 560-577, 605-621)

1b) In addition, the manuscript would improve if a final figure summarizing the main events in translational control during health and neurodevelopmental diseases was added.

We have decided to not include such a figure because there are several reviews by us and others on this topic, while the focus of this paper is the neurovascular interplay. Instead, we have prepared a table describing the link between translational control and NDDs with references to the respective reviews.

2) I suggest authors include a final section of Discussion and perspectives in the field, especially concerning translational control of neurovasculature in neurodevelopmental diseases, as this seems to be a fertile research ground.

We feel that the addition of such a section would be repetitive and summarizing several key facts already expanded in the text. The current last paragraph, which comes at the end of the “**Translational control of neurovascular development and neurodevelopmental disorders**” section, contains perspectives that are pertinent to the title of this review and the neuro-vascular interplay. Furthermore, we and others have recently published reviews on future perspectives of translational control in brain (Aguilar-Valles et al., 2018; Atlasi et al., 2020; Jung et al., 2014; Knight et al., 2020; Wiebe et al., 2020).

Aguilar-Valles, A., Matta-Camacho, E., & Sonenberg, N. (2018). Translational Control Through the eIF4E Binding Proteins in the Brain. In W. Sossin (Ed.), *The Oxford Handbook of Neuronal Protein Synthesis*. Oxford University Press. <https://doi.org/10.1093/oxfordhb/9780190686307.013.5>

Atlasi, Y., Jafarnejad, S. M., Gkogkas, C. G., Vermeulen, M., Sonenberg, N., & Stunnenberg, H. G. (2020). The translational landscape of ground state pluripotency. *Nature Communications, 11*(1). <https://doi.org/10.1038/s41467-020-15449-9>

Jung, H., Gkogkas, C. G., Sonenberg, N., & Holt, C. E. (2014). Remote control of gene function by local translation. *Cell, 157*(1), 26–40. <https://doi.org/10.1016/j.cell.2014.03.005>

Knight, J. R. P., Garland, G., Poyry, T., Mead, E., Vlahov, N., Sfakianos, A., Grosso, S., DeLima-Hedayioglou, F., Mallucci, G. R., Von Der Haar, T., Smales, C. M., Sansom, O. J., & Willis, A. E. (2020). Control of translation elongation in health and disease. *DMM Disease Models and Mechanisms, 13*(3). <https://doi.org/10.1242/dmm.043208>

Wiebe, S., Nagpal, A., & Sonenberg, N. (2020). Dysregulated translational control in brain disorders : from genes to behavior. *Current Opinion in Genetics & Development, 65*, 34–41. <https://doi.org/10.1016/j.gde.2020.05.005>